# Beyond Faithfulness: A Framework to Characterize and Compare Saliency Methods

## Abstract

Saliency methods calculate how important each input feature is to a machine learning model's prediction, and are commonly used to understand model reasoning. "Faithfulness," or how fully and accurately the saliency output reflects the underlying model, is an oft-cited desideratum for these methods. However, explanation methods must necessarily sacrifice certain information in service of user-oriented goals such as simplicity. To that end, and akin to performance metrics, we frame saliency methods as *abstractions*: individual tools that provide insight into specific aspects of model behavior and entail tradeoffs. Using this framing, we describe a framework of nine dimensions to characterize and compare the properties of saliency methods. We group these dimensions into three categories that map to different phases of the interpretation process: *methodology*, or how the saliency is calculated; *sensitivity*, or relationships between the saliency result and the underlying model or input; and, *perceptibility*, or how a user interprets the result. As we show, these dimensions give us a granular vocabulary for describing and comparing saliency methods — for instance, allowing us to develop "saliency cards" as a form of documentation, or helping downstream users understand tradeoffs and choose a method for a particular use case. Moreover, by situating existing saliency methods within this framework, we identify opportunities for future work, including filling gaps in the landscape and developing new evaluation metrics.

## 1 Introduction

As machine learning (ML) systems are increasingly deployed into real-world contexts, stakeholder interviews (Tonekaboni et al., 2019; Bhatt et al., 2020), design best practices (Amershi et al., 2019), and legal frameworks (European Commission, 2018) have underscored the need for explainability. Saliency methods — a class of attribution methods that aim to identify features in an input that were important to a trained ML prediction — are frequently used to provide explanations. However, each method operates differently and, thus, multiple methods can produce seemingly varying explanations for the same model and input. How, then, should we reason about choosing and comparing methods for a particular application?

Prior work has suggested that "faithfulness," or how accurately a saliency result reflects the underlying model, is a desideratum for these methods (Li et al., 2021; Ding & Koehn, 2021; Tomsett et al., 2020; Adebayo et al., 2018). In this paper, however, we argue that faithfulness is not a productive goal for saliency methods — by design, they cannot offer a complete and accurate reflection of a model's behavior, akin to a printout of model weights. Rather, we frame saliency methods as *abstractions* of model behavior that selectively preserve and necessarily sacrifice information in service of human-centric goals such as simplicity and understandability.

With this framing, we propose a nine-dimensional framework to characterize and compare saliency methods. These dimensions fall into three categories, corresponding to different parts of the interpretation process: *methodology*, or how the saliency is computed; *sensitivity*, or relationships between the saliency and the underlying model or input; and *perceptibility*, or how an end-user perceives the saliency. These dimensions decompose a singular notion of faithfulness into more granular units that can be reasoned about individually: by situating methods along these dimensions, we can surface their relative strengths, limitations, and differences to better understand trade offs that are latent in their design. In doing so, we demonstrate how our framework allows us to develop "saliency

| Method | Description |
|---|---|
| **Vanilla Gradients (VG)** (Erhan et al., 2009; Simonyan et al., 2013) | Computes the gradient of the model's output for the class of interest with respect to the input. |
| **Input × Gradient** | Extends VG by performing element-wise multiplication on the gradient and input features. |
| **Integrated Gradients (IG)** (Sundararajan et al., 2017) | Interpolates between a baseline example and the input, accumulating gradients at each step. |
| **GradCAM** (Selvaraju et al., 2017) | Computes the gradient of the model's output for the class of interest with respect to the last convolutional layer of a CNN. |
| **SmoothGrad** (Smilkov et al., 2017) | Averages over saliency results for slightly perturbed/noisy versions of the input. |
| **Guided Backpropagation (Guided BP)** (Springenberg et al., 2014) | Extends gradient-based methods by preventing the flow of negative gradients, resulting in attributions from positive paths in the network. |
| **Local Interpretable Model-Agnostic Explanations (LIME)** (Ribeiro et al., 2016) | Trains a simple surrogate model on a localized dataset generated by perturbing the input example, and uses its coefficients as importance values. |
| **Shapley Additive Explanations (SHAP)** (Lundberg & Lee, 2017) | Computes attribution as a game theory problem where each feature is a "player" and the output of the model is a "payout" distributed amongst them. |
| **Meaningful Perturbations (MP)** (Fong & Vedaldi, 2017) | Strategically masks different parts of the input to learn which are important (i.e., lead to the largest change in the output). |
| **Sufficient Input Subsets (SIS)** (Carter et al., 2019) | Uses instance-wise backward selection to identify subsets of input features sufficient for the model to make its prediction above some confidence threshold. |
| **RISE** (Petsiuk et al., 2018) | Computes the weighted sum of the model's output on masked versions of the input. |
| **XRAI** (Kapishnikov et al., 2019) | Identifies important image regions by segmenting the image into many overlapping regions and ranking the regions based on the sum of pixel-attribution within each region. |

Table 1: Summary of saliency methods discussed throughout the framework.

cards" (akin to "model cards" (Mitchell et al., 2019) and "datasheets" (Gebru et al., 2018)) to document individual methods and to better contextualize saliency results. Moreover, using a concrete example of ML-based radiology diagnostic systems, we show how downstream stakeholders can use our framework to weigh tradeoffs and choose a task-appropriate method. Finally, we show how our framework identifies compelling opportunities for future work including exploring understudied dimensions and developing new metrics that target particular dimensions.

## 2    RELATED WORK: SALIENCY METHODS AND THEIR EVALUATIONS

Saliency methods (sometimes referred to as feature attribution methods) produce explanations for an ML model's output. Given an input, saliency methods compute an importance score for each input feature describing its influence on the model's output. Existing categorizations for saliency methods have focused on algorithmic properties (gradient or perturbation-based, path-attribution or gradient-only) Molnar (2019). Our framework aims to categorize a broader range of important characteristics. We apply our framework to a variety of common saliency methods listed in Table 1.

Evaluations of saliency methods have primarily focused on how accurately their results represent model behavior, often referred to as *faithfulness*. A growing body of work has identified failures of some methods such as susceptibility to adversarial perturbations (Ghorbani et al., 2019), lack of neuron discriminativity (Mahendran & Vedaldi, 2016), and a predisposition towards input recovery (Nie et al., 2018; Adebayo et al., 2018). Other work has proposed proxy tests that measure different aspects of faithfulness. Adebayo et al. (2018) recommend model randomization and data label randomization tests, and Kindermans et al. (2019) test whether constant input shifts affect saliency results. Samek et al. (2016) judge saliency methods by iteratively replacing features that have high importance values with random noise and measuring how much the output changes. While these tests quantitatively analyze saliency methods, Tomsett et al. (2020) found they can produce inconsistent rankings.

An alternate line of work has looked to break faithfulness down into measurable axioms. Sundararajan et al. (2017) propose implementation invariance (a saliency method should produce the same output on functionally equivalent models), sensitivity (a saliency method should give importance to a feature if and only if changing it leads to a different output), and linearity (if a model is the composition of two sub-models, the saliency method's output for the model should be the weighted

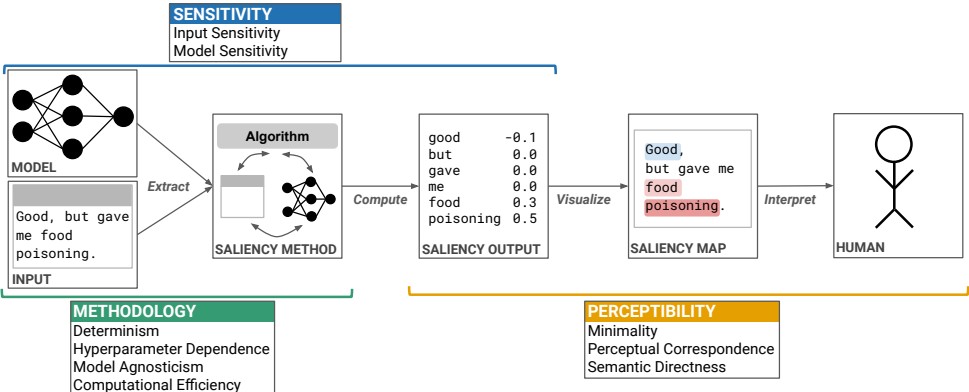

Figure 1: Our framework offers nine dimensions to characterize and compare saliency methods, which are grouped into three categories corresponding to different phases of the interpretation process: *methodology* describes how the saliency is calculated; *sensitivity* describes relationships between input and output; and *perceptibility* relates to human interpretation.

sum of its outputs for each sub-model). Sundararajan et al. (2017) and Shrikumar et al. (2017) both posit a saliency method should exhibit completeness — a saliency method's attributions should sum to the difference between the model's output on the input and the model's output on a neutral input. Shrikumar et al. (2017) and Montavon et al. (2018) claim saliency methods should output a continuous function. Montavon et al. (2018) also propose the axiom of selectivity — a saliency method should distribute importance to features that have the greatest impact on the model's output. Finally, Kindermans et al. (2019) state that a saliency method should be invariant to constant transforms.

While axioms are presented as constraints that all saliency methods should attain, the axes in our framework describe attributes of saliency methods that can be traded off to select the best method for different use cases. To situate saliency methods along each axis, we utilize existing tests (Kindermans et al., 2019; Adebayo et al., 2018); however, our framework surfaces the need for additional evaluations, since existing tests do not fully describe each axis and have not assessed all methods.

## 3 NINE DIMENSIONS TO CHARACTERIZE & COMPARE SALIENCY METHODS

To distill a language to characterize and compare saliency methods, we treat saliency methods as *abstractions* of the underlying model behavior. That is, to explain model behavior in a human-understandable format (e.g., a feature attribution heatmap), we must necessarily abstract away some detail and concreteness. A saliency method, by design, is not as precise a reflection of model behavior as a printout of the weights, but it is much more human-interpretable. The idea of abstraction recurs in other parts of the machine learning pipeline. For example, "test accuracy" is an abstraction of model performance: it does not capture all aspects of model performance but is nevertheless a convenient approximation. Abstractions selectively preserve information and can be combined to arrive at a more complete picture of the underlying phenomenon — i.e., combining test accuracy with AUROC to more fully understand model performance. This framing reveals that a single saliency method cannot fully explain model behavior. Instead, it is necessary to understand different methods' strengths and limitations to choose an appropriate method for the model, domain, and task.

Our framework defines nine dimensions that describe *what* information saliency methods abstract, and *how* this abstraction is computed. We group these dimensions into three categories that map to different stages of the interpretability process (see Figure 1). *Methodology* covers the model, input, and saliency method, and dimensions under this category describe how the saliency method operates. *Sensitivity* goes one stage further, with dimensions that describe the relationship between the saliency output and the model or input. *Perceptibility* covers the final stages of the process, with dimensions that describe how users perceive the saliency output and any associated visualizations. These dimensions allow us to decompose faithfulness into aspects that can be reasoned about individually. For instance, Figure 2 demonstrates how our dimensions allow us to situate saliency methods in relation to one another to understand their strengths, limitations, and differences. In the following subsections, we describe and provide an illustrative example of each dimension.

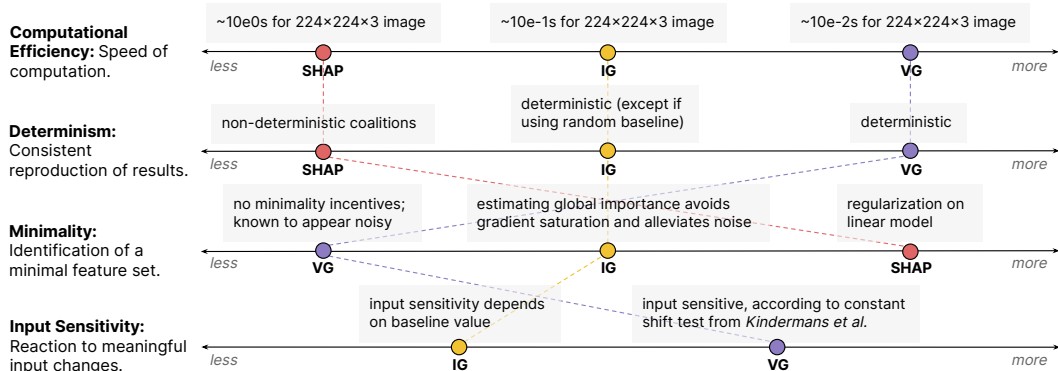

Figure 2: Situating methods along dimensions of our framework reveals their multifaceted differences (e.g., SHAP is more minimal but less tractable than VG). Understanding these differences can inform context-specific decisions about how to choose and interpret methods appropriately. It also reveals gaps (e.g., SHAP has not been tested for input sensitivity) that can inform future work.

## 3.1 METHODOLOGY: HOW SALIENCY METHODS OPERATE

**Determinism** Some saliency techniques are non-deterministic, and running them with different random seeds can produce significantly different outputs. For instance, methods such as RISE or SHAP rely on randomly-generated masks or coalitions. The output from a non-deterministic method represents only one example from a potentially high-variance distribution, and may be afforded authority that does not account for its inherent uncertainty. Although we may never explicitly choose a method for its non-determinism, we may nevertheless accept non-determinism in service of other dimensions such as increased model agnosticism (e.g., RISE), perceptual correspondence (e.g., SHAP), or minimality (e.g., MP).

*Example.* Consider a model assisting dermatologists diagnosing skin cancer. In Figure 3(a), we show that two runs of a non-deterministic method (LIME) on a melanoma prediction model result in different saliency maps. These variations may have significant consequences given small areas of the lesion or surrounding skin may be integral to the diagnosis. Visualizing and interpreting multiple runs together is time-consuming and confusing, but only looking at one run could skew a clinician's judgment. Thus, in this case, it might make sense to prioritize a method with deterministic outputs.

**Hyperparameter Dependence** Hyperparameter dependence captures how many hyperparameters or design decisions the user must specify to run a particular method, and how sensitive it is to them. If there are not sufficient resources or expertise to devote to hyperparameter optimization, simply using default values can lead to misleading results. Similarly, misleading or confusing results might arise if the hyperparameters were chosen based on a particular dataset but deployed in a setting in which there is a significant distribution shift. In situations like this, it makes sense to prioritize methods with low hyperparameter dependence. In other cases, where developers have dedicated appropriate time and resources to tuning, it might be preferable to use a method dependent on hyperparameters (e.g., IG, SmoothGrad) because it improves other dimensions (e.g., minimality).

*Example.* IG computes feature importances by interpolating between a "meaningless" baseline input and the actual input, accumulating the gradients at each step. A common practice is to use a baseline value of all zeroes; however, in some cases, a zero baseline can be misleading and potentially harmful. Take a model trained to predict bone fractures from x-ray images. Fractures often appear as a black line in the bone, but, since they have the same pixel value as the baseline (zero), the IG result will indicate that they are not important. In Figure 3(b), we show a similar example where the choice of baseline (black, white) has a strong effect on the saliency map. Choosing an appropriate baseline requires time and a deep understanding of the data and method; if this is not available in a particular application, it is better to prioritize a saliency method without important hyperparameters.

**Model Agnosticism** Model agnostic methods (i.e., SHAP) treat the underlying model as a black-box, relying only on its input and output. On the other hand, model-dependent methods need access to various model internals, and may have specific requirements, such as differentiability (i.e.,

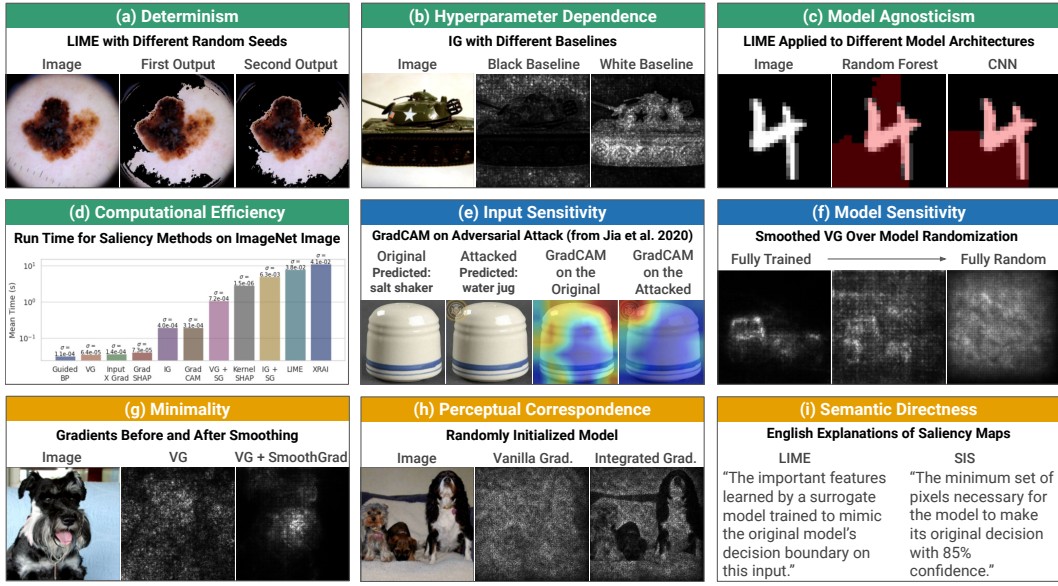

Figure 3: Our framework defines nine dimensions of saliency methods, shown above with examples. Each dimension describes an attribute of saliency methods that can be reasoned about and traded off.

gradient-based methods) or a particular type of architecture (i.e., GradCAM requires a CNN). Model agnosticism is a priority for use cases when modularity is required, when ensembling models, when comparing across model architectures, or when the model internals are not accessible. However, if only considering a specific architecture, one might opt for a non-model-agnostic method like Grad-CAM that provides other benefits (i.e., increased model sensitivity).

*Example.* Saliency methods can be an informative tool to compare the behavior of different models on the same input. For a direct comparison, the same saliency method should be used across models, and a model agnostic method is likely necessary. For example, in Figure 3(c), we show how a model-agnostic method like LIME can produce explanations for a CNN and a random forest model, which would not be possible using a model-dependent method such as IG or GradCAM.

**Computational Efficiency** Different methods vary widely in their computational efficiency — for example, perturbation-based methods (e.g., SIS, MP) are often more computationally intensive than simple gradient-based methods (e.g., VG and its variants). In a time-sensitive, low-resource, or data-intensive setting (e.g., all of ImageNet), prioritizing computational simplicity (which may entail sacrificing along other axes) may be necessary.

*Example.* Video modeling (i.e., action recognition) is computationally-intensive because, unlike 3-dimensional image inputs (channels x height x width), video inputs add a fourth dimension (frames). Even though the original frame rate can be subsampled, the compute time required for frame-wise saliency maps scales linearly with video length. As shown in Figure 3(d), methods like Guided BP are orders of magnitude faster than other methods like XRAI. Thus, in the video modeling setting, choosing a computationally efficient saliency method may be necessary to interpret model behavior.

## 3.2 SENSITIVITY: RELATIONSHIPS BETWEEN SALIENCY OUTPUT AND THE MODEL/INPUT

**Input Sensitivity** Input sensitivity captures the concept that the result of a saliency method should reflect the model's sensitivity to transformations in input space. Prior work has attempted to quantify the input sensitivity of saliency methods by comparing results before and after applying a constant shift to the input (Kindermans et al., 2019) (i.e., positing that an input shift that does not change the model output should also not change the saliency map if the method is input sensitive). Other work defines the property such that if two inputs differ by only one feature but result in different model outputs, an input-sensitive saliency method should assign a non-zero value to that feature. In some cases, we might tradeoff input sensitivity for, e.g., minimality (as in SmoothGrad).

*Example.* Input sensitivity is particularly critical for applications that risk adversarial attacks or when studying model behavior in the presence of these attacks. For example, Adv-watermark (Jia et al., 2020) adds watermarks to images that change the model's prediction. In Figure 3(e), we show an example from Jia et al. (2020) where the model correctly predicts salt shaker on the original image but, after the addition of a watermark, predicts water jug. The GradCAM saliency is input sensitive and identifies the adversarial watermark as most important to the incorrect water jug prediction.

**Model Sensitivity** Model sensitivity is the concept that the result of a saliency method should reflect changes in the model parameters. Prior work has formalized notions of model sensitivity in different ways: for example, Adebayo et al. (2018) run a parameter randomization test (comparing the output of a saliency method on the trained model and a randomly-initialized model), and Sundararajan et al. (2017) define an implementation invariance axiom (stating that a saliency method should produce consistent outputs on functionally equivalent models). In some cases, we might be okay with trading-off model sensitivity. For example, Guided BP is not model sensitive, but is input sensitive; if we cared more about examining changes across different inputs (as opposed to different models), we might choose it over a more model sensitive method.

*Example.* Prioritizing model sensitivity may be particularly important when comparing models. In particular, take the case where one compares a model to its fine-tuned counterpart to understand the effect on the model's behavior. In this case, it is critical to have a method with model sensitivity; otherwise, the results might misleadingly indicate that the model has not changed. In Figure 3(f), we use VG, a model-sensitive method, showing that it is sensitive to cascading layer randomization.

### 3.3 PERCEPTIBILITY: HUMAN INTERPRETATION OF SALIENCY OUTPUT

**Minimality** Minimality refers to how many features are given a significant value in the saliency map. Methods that attribute importance to many input features, especially when applied to high-dimensional data, can produce results that are difficult to interpret. For example, several papers have observed that VG results appear noisy, sometimes giving almost every pixel in an image a non-zero importance. On the other hand, methods like LIME and SIS directly incorporate minimality into their procedures using regularization and backward selection, respectively. Smoothing methods, such as SmoothGrad, also improve minimality by averaging over the results on slight perturbations of the input. At the same time, methods that do not prioritize minimality may have other benefits (e.g., VG are model sensitive) and might be more appropriate depending on the context.

*Example.* Minimality is particularly important when interpreting high-dimensional data. For example, when dealing with patient medical records, each record might contain hundreds or thousands of features. Without minimality, the resulting saliency values risk obscuring whatever signal is present and increasing the cognitive load required to interpret the results. In Figure 3(g), we compare results of a saliency method that are not minimal (VG) before and after smoothing (SmoothGrad) is applied, demonstrating that minimality makes it significantly easier to glance at and interpret results.

**Perceptual Correspondence** Perceptual correspondence captures the idea that the perceived signal in the saliency result should reflect the model's confidence. For example, if a model is randomly guessing predictions, saliency values should appear correspondingly random. Perceptual correspondence is crucial for high-risk settings, where a misleading signal could provide an unwarranted justification for essential decisions or lead users down incorrect paths. On the other hand, if saliency results are not interpreted individually for decision-making, but rather being used for larger-scale analyses of model behavior that will aggregate one-off artifacts (e.g., Boggust et al. (2021) use aggregate metrics on saliency results across an entire dataset to measure human-AI alignment), we might prioritize low computational efficiency over perceptual correspondence.

*Example.* In Figure 3(h), we show saliency maps on a randomly initialized model. While VG appears random, IG computed with a black baseline exhibits a perceptual signal because of the near-black regions in the images. A user might project meaning onto the saliency map, concluding that the model is picking up on signals in the input when in fact it is entirely random. While we use an ImageNet image (Deng et al., 2009) to illustrate the concept, the lack of perceptual correspondence in IG would be particularly problematic in black and white medical images (i.e., x-rays, mammograms) where saliency maps might indiscriminately highlight white areas, providing unwarranted justification for further action or treatments.

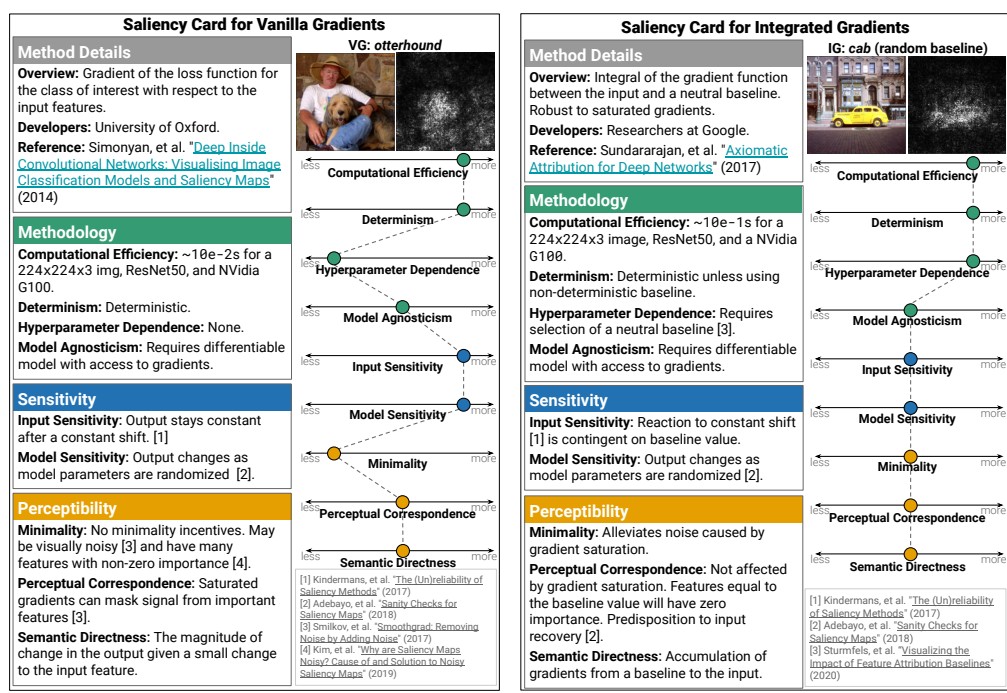

Figure 4: Using our framework, individual methods can be documented in a standardized manner as "saliency cards," informing users about tradeoffs and limitations, and facilitating rapid comparison.

**Semantic Directness** Saliency methods represent different aspects of model behavior. For example, the result from SIS represents minimal sets of pixels necessary for a particular prediction; the result from LIME, on the other hand, represents the learned coefficients of a local surrogate model. Semantic directness refers to how straightforward it is for a particular user group to understand what the result of a saliency method represents. A semantically direct method may be particularly important if users are unfamiliar with ML or saliency methods, in order to prevent misinterpretation of results. In other cases, we might trade it off — for example, while SHAP (which defines feature importances as Shapley values, a method from game theory) may not be semantically direct to many users, it can improve perceptual correspondence, which might be more critical in high-risk contexts.

*Example.* Recently, a range of public-facing browser-based interpretability tools have been developed, such as the Language Interpretability Tool (LIT) (Tenney et al., 2020). These tools might be helpful for data scientists or other users in a particular domain, who don't necessarily have formal ML expertise, to explore their data and a model's behavior on it. Within LIT, users can choose among several methods for viewing the saliency results of a model on selected sentences. In this case, a non-expert user might opt for a more semantically direct method such as SIS (over, e.g., IG, SHAP, or LIME) that facilitates more intuitive and quick exploratory analyses without needing to understand the implications of a surrogate model or accumulated gradients.

## 4 FRAMEWORK USE CASES

### 4.1 DOCUMENTING SALIENCY METHODS WITH "SALIENCY CARDS"

By moving beyond a singular notion of faithfulness, our framework gives us a rich, granular vocabulary to describe existing and future methods, and situate them in relation to one another. In doing so, our framework improves our understanding of different methods and can be used to document them in a standardized fashion. For example, in the same way "model cards" and "datasheets" provide detailed, public-facing documentation about particular models or datasets, our dimensions can inform an analogous "saliency card" as shown in Figure 4. Such cards step through the individual dimensions of the framework, providing a brief qualitative or quantitative description for each. By including an example of the method in action, as well as a schematic figure that plots the method

with respect to each dimension, these saliency cards can help better contextualize individual methods in the broader landscape. Downstream users can rapidly compare cards to understand a method's strengths and limitations, as well as build intuition for the trade offs between different methods.

## 4.2 COMPARING AND CHOOSING SALIENCY METHODS IN A RADIOLOGY SETTING

The dimensions of our framework let us systematically evaluate, choose, and compose methods for different contexts. To illustrate this use case, we consider an application of ML-based diagnostic tools: interpreting chest x-rays. Radiologists interpret chest x-rays to diagnose a range of cardiopulmonary conditions, and automated analysis could potentially improve their workflows and extend expertise to low-resource settings. ML-based systems have gained substantial traction for this application, in part because of the availability of large, public datasets (Johnson et al., 2019; Irvin et al., 2019). In practice, accompanying automated decisions with explanations is critical, both for the systems to be usable and trusted by physicians (Tonekaboni et al., 2019) and to follow standards and laws for explainability (Smith, 2020; European Commission, 2018).

We evaluate our framework in the context of this case study through semi-structured interviews with 4 radiologists (see Section A.1 for interview details) to understand how our framework surfaces trade offs and helps radiologists compare different saliency methods. We find radiologists may be unfamiliar with the the way non-determinism manifests in saliency methods (i.e., LIME, SHAP), given that it is typically not encountered within other medical technologies. Non-deterministic results could appear random and lead to speculation about whether an important feature would still be important if they reran the method. Given this, we may want to prioritize determinism over other axes and choose a method, like VG, that is fully deterministic. While VG has less minimality than many non-deterministic methods, its noisiness may be preferable to the stochasticity of a non-deterministic method — background noise is a familiar concept in medical imaging and radiologists are used to accounting for it during interpretation.

Relatedly, minimality is often presented as a desirable attribute (Smilkov et al., 2017). Radiologists, however, are accustomed to using background noise to attenuate measurement uncertainty in their work; thus, the absence of background noise could convey unwarranted certainty. Indeed, results in visualization research indicate that clean, minimalist visual representations can convey a misplaced sense of certitude about the underlying data (Kennedy et al., 2016; Kostelnick, 2008). This consideration also suggests another tradeoff — between perceptual correspondence and minimality. If the saliency result misleadingly communicates a signal to which a radiologist attributes meaning, it could lead to a patient being misdiagnosed or wrongly treated. Given the high risk involved with incorrect decisions in a medical context, we might actually opt for a *less* minimal method — despite the fact that its results may not be as glanceable — in order to increase perceptual correspondence.

Semantic directness is similarly presented as an beneficial characteristic for methods used with non-ML-experts (Carter et al., 2019). Radiologists, however, are accustomed to reading and interpreting scans produced via magnetic resonance imaging (MRI) technology despite having a limited technical understanding of the physics underlying MRI technology. A similar ethos likely applies to saliency methods as well — radiologists might be comfortable receiving saliency results even without understanding the underlying mechanisms (e.g., gradients, neural network architectures) as long as they know how to use them.

The deployment process for a clinical decision model might also have implications on which dimensions to prioritize. For example, IG's hyperparameter dependence could have significant consequences for x-ray data (e.g., the commonly used black baseline would lead to misleading results for x-ray images). This dependence might be more or less of an issue depending on the expertise of the person tuning the hyperparameters. If tuning was left to the software vendor, for example, we might prioritize a method that does *not* require consequential decisions like IG's baseline value.

Overall, the dimensions of our framework provide valuable structure for probing an application and reasoning about tradeoffs. While previously, we might have picked a method to use more arbitrarily, we can now systematically think about attributes to target. For example, non-determinism and perceptual correspondence are likely necessary for a radiology context, while minimality and semantic directness are not. Based on this, we can see how a method like VG might be better suited than LIME for this particular task. Beyond specific methods, we can gain insight into general design implications, such as whether hyperparameters should be set beforehand or open to user choice.

### 4.3 IDENTIFYING NEW RESEARCH DIRECTIONS

Our framework also allows us to systematically analyze the research landscape. We characterize 10 methods along each of our framework's dimensions (Table A1) by drawing on existing work (i.e., papers that present or compare methods) and our experiments (i.e., running methods locally to compare their computational burden). In doing so, we identify several patterns and gaps in existing methods that suggest promising directions for future work to explore.

Some dimensions are rarely explored in existing work. For example, while it may be a crucial attribute for high-risk applications, few papers attempt to measure perceptual correspondence. This gap might be due to measurement difficulty, since research characterizing and quantifying perceptual correspondence requires conducting user studies. Such inquiries could draw from work on human visual perception (Logothetis & Sheinberg, 1996) — for example, using eye-tracking to measure if and where in an image people fixate (Borys & Plechawska-Wójcik, 2017; Yarbus, 2013) — to designing standardized ways of studying when people read strong signal in saliency map visualizations. A better understanding of perceptual correspondence could also inform the design of saliency visualizations that explicitly communicate limitations of the method and preemptively avoid implying unwarranted signal. For instance, we can imagine a richer design space beyond static heatmaps where interaction is used to dynamically overlay multiple attributions (as in Olah et al. (2018)) and more intuitively communicate uncertainty.

In other cases, existing work has already begun to devise evaluative metrics that map to dimensions in our framework. For example, Adebayo et al. (2018) use a model parameter randomization test that helps gauge model sensitivity, and Kindermans et al. (2019) use a test that compares saliency results before and after applying a constant shift that helps measure input sensitivity. However, these are specific, narrow ways of measuring model and input sensitivity, and can produce inconsistent results (Tomsett et al., 2020). By explicitly identifying these properties as separate dimensions, there is significant opportunity to devise new or different ways to quantify them. For example, while Kindermans et al.'s quantification of input sensitivity involves applying a meaning*less* change to the input, we could imagine applying a meaning*ful* change to the input and studying how the output changes in response (akin to algebraic visualization design by Kindlmann & Scheidegger (2014)).

Once methods are well-characterized across a particular dimension, other patterns might emerge — for example, clustering within a dimension or combinations of dimensions. We noticed, for instance, that minimality is typically encouraged via regularization or smoothing terms, and thus the more minimal methods are typically more hyperparameter dependent. Are these dimensions inherently tied together, or is it possible to achieve increased minimality in other ways? This is an example of a research question that emerges from understanding gaps and clusters in the design space.

## 5 CONCLUSIONS AND FUTURE WORK

In this paper, we present a framework for describing, communicating, and comparing saliency methods. Unlike prior approaches, which have focused on a singular notion of "faithfulness" or how accurately saliency output reflects the underlying model, our framework offers nine dimensions to characterize and compare saliency methods that span different phases of the interpretation workflow. As a result, our framework facilitates rich documentation of saliency methods (i.e., through saliency cards), and can help users understand trade offs and select methods that are appropriate for their data, model, and task. Finally, situating saliency methods along each dimension surfaces areas for future research including developing methods that prioritize specific attributes, designing tests to quantify each dimension, and proving (or disproving) entanglement between dimensions.

We intend our framework to start a conversation around approaches to characterize and compare saliency methods, and to be a "living artifact." Namely, as researchers develop new saliency methods or quantify particular dimensions (e.g., input sensitivity), we expect our framework to evolve. For instance, additional dimensions may be introduced to distinguish attributes or techniques proposed by novel methods, or new evaluative studies may reveal that existing dimensions are too broadly defined and warrant further decomposition into constituent, testable axes. Further work is also needed to provide a formal (i.e., mathematical) treatment of our framework. Doing so would help turn our framework from a purely descriptive instrument into a generative one as well — where dimensions can be operationalized to systematically produce altogether new saliency methods.

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

# A  APPENDIX

| | Determinism | Hyperparameter Dependence | Model Agnosticism | Computational Efficiency | Input Sensitivity | Model Sensitivity | Minimality | Perceptual Correspondence | Semantic Directness |
|---|---|---|---|---|---|---|---|---|---|
| **VG** | Deterministic. | None. | Requires a differentiable model and access to gradients. | ~10e-2 seconds for a 224x224x3 image. | Saliency results stay constant after a constant input shift. | Saliency results change as model parameters are randomized. | Results appear noisy if gradients for important features are saturated and normalized within an example. | Saturated gradients can mask signal from important features. | The magnitude of change in the output given a small change in an input feature. |
| **Input X Gradient** | Deterministic. | None. | Requires a differentiable model and access to gradients. | ~10e-2 seconds for a 224x224x3 image. | Saliency results change after a constant input shift. | Saliency results change as model parameters are randomized. | Results are more minimal than VG since only features with high gradients and input values appear important. | Predisposition to input recovery; susceptible to gradient saturation. | The input feature value weighted by the gradient. |
| **IG** | Deterministic unless using a random baseline. | Baseline value; number of steps to estimate the integral. | Requires a differentiable model and access to gradients. | ~10e-1 seconds for a 224x224x3 image. | Reaction to constant shift test dependent on baseline value. | Saliency results change as model parameters are randomized. | Can alleviate noisiness from gradient saturation by estimating global importances. | Predisposition to input recovery (can be exacerbated by choice of baseline). Addresses gradient saturation problems. | The accumulated gradient between the baseline value and and feature value. |
| **GradCAM** | Deterministic. | Interpolation method to upsample to the original feature space. Choice of convolutional layer (typically last convolutional layer). | Requires a differentiable model, access to gradients, and convolutional layers. | ~10e-1 seconds for a 224x224x3 image. | | Saliency results change as model parameters are randomized. | Interpolating back to original input size produces a low-resolution result. | | The positive attributions of the gradient-weighted feature maps from the last convolutional layer. |
| **SmoothGrad** | Non-deterministic perturbations. | Gaussian noise parameters; number of runs to average over. | Can be applied to any saliency method. | Adds ~10x time increase for 224x224x3 image. | Reaction to constant shift test inherited from underlying saliency method. | Reaction to model parameter randomization inherited from underlying saliency method. | Minimality encouraged by averaging over multiple results. | | The average of many runs of another saliency method on perturbed versions of the input. |
| **Guided Backprop** | Deterministic. | None. | Requires a differentiable model and access to gradients. | ~10e-2 seconds for a 224x224x3 image. | Saliency results stay constant after a constant input shift. | Saliency results do not change when higher layer weights are randomized, and only change when lower layer weights are randomized. | Minimality encouraged through removing negative gradients. | Predisposition to input recovery. | The output of another gradient-based saliency method (typically VG), only considering paths through the network with positive gradients. |
| **LIME** | Non-deterministic perturbations. | Linear surrogate models to search over; input perturbation parameters; linear model parameterization. | No requirements on the model or access to internals needed. | ~10e0 seconds for a 224x224x3 image. | | | Minimality encouraged through regularization in local linear model. | | The positively contributing features learned by a surrogate model trained to mimic the original model's local decision boundary for a particular input. |
| **SHAP** | Non-deterministic coalition sampling. | Feature replacement values; linear model parameterization; regularization parameter. | No requirements on the model or access to internals needed. | GradSHAP: ~10e-2 seconds for a 224x224x3 image. KernelSHAP: ~10e0 seconds for a 224x224x3 image. | | | Minimality encouraged through regularization in local linear model. | Features with no impact on model given zero importance in result. Features that impact the model equally receive equal attribution. | The impact of each input feature on the output as defined by Shapley values. |
| **MP** | Masks pixels with non-deterministic perturbations. | Regularization parameter; noise and blur parameters. | No requirements on the model or access to internals needed. | | | | Minimality encouraged through regularization on the meta-predictor that learns explanatory rules. | | The minimal region of the image that would cause the largest change if removed. |
| **SIS** | Deterministicly produces a set of explanations per input. | Feature replacement values; model confidence threshold. | No requirements on the model or access to internals needed. | Algorithm described in Carter et al. (2019) is prohibitively slow on 224x224x3 images. | | | Selects a minimal set of features sufficient to meet the confidence threshold. | | The minimum set of pixels necessary for the model to produce the output with probability above a given threshold. |
| **RISE** | Non-deterministic mask generation. | Masking value; mask generation parameters. | No requirements on the model or access to internals needed. | ~10e-1 seconds for a 224x224x3 image. | | | Mask upsampling encourages continuous salient regions. | | The weighted sum of the model's confidence on masked versions of the input for inputs that include that feature. |
| **XRAI** | Deterministic. | Segmentation method; attribution method (paper proposed IG with black and white baselines). | Requires a model whose inputs features can be meaningfully clustered together (e.g., image pixels). | ~10e1 seconds for a 224x224x3 image. | | | Outputs region-level importance. | Region-importance has been shown to be more human interpretable than pixel-importance. (Sundararajan et al., 2019) | The regions with the largest sum of feature attribution. |

Table A1: We situate 10 saliency methods along the nine dimensions of our framework, finding that they capture meaningful and distinct attributes about the methods. Some dimensions are characterized via our own experiments (e.g., computational efficiency) and others draw from tests described in existing work (e.g., model and input sensitivity). In this way, our framework lets us systematically analyze the research landscape to understand gaps and opportunities for future work.

## A.1 RADIOLOGIST INTERVIEW DETAILS

To evaluate the effectiveness of our framework, we interviewed 4 radiologists and report the results in Section 4.2. We recruited radiologists via a colleague who is a practicing radiologist at a hospital. We did not require radiologists to have any familiarity or experience with machine learning. Each interview lasted approximately 30 minutes. During the interview we discussed dimensions of our framework and gave clinical examples on CheXpert (Irvin et al., 2019) chest x-rays (see Figure A5). We asked radiologists open-ended questions with the goal of understanding:

- How do radiologists think about individual dimensions of the framework?
- How do radiologists prioritize and trade off different dimensions?
- How well does the framework enable effective communication about saliency methods between the radiologists and ourselves?

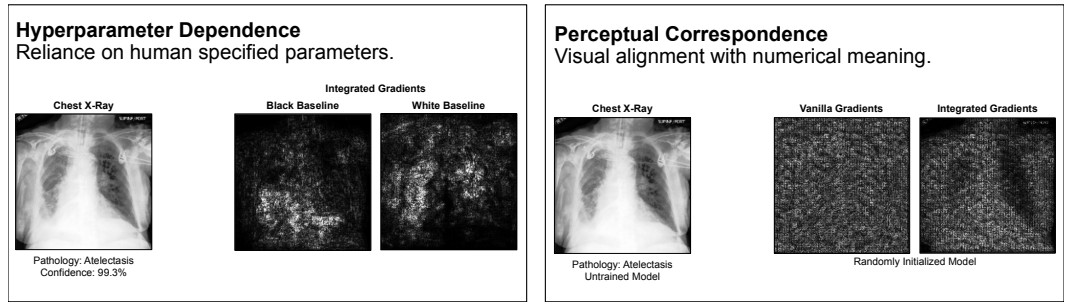

Figure A5: Chest x-ray examples shown to radiologists in our interviews.

