# OpenReview forum: "Beyond Faithfulness: A Framework to Characterize and Compare Saliency Methods"
_ICLR.cc/2022/Conference — ICLR 2022 Submitted_

### Official Review · Reviewer_DxGz · 2021-11-02

**Correctness:** 3
**Technical Novelty And Significance:** 3
**Empirical Novelty And Significance:** 3
**Recommendation:** 5
**Confidence:** 5

**Main Review:**

The paper is entirely a survey/review of existing methods; it does not propose any new XAI algorithms. However, I found the proposed dimensions and comparison of the saliency methods along these dimensions very interesting and informative. I could see this framework being useful to researchers in XAI as well as end users. The paper is also very well written and clear.

The main weakness of the paper in my opinion are:

(1) The methodology is not entirely transparent -- it's not clear how the models are ranked on these dimensions or how a "score" would be computed for a given dimension. The authors note that some dimensions like "perceptual correspondence" are difficult to measure and may require large-scale user studies; other dimensions may be measured in multiple different ways and different methods could give different ranking results. It's not clear how this would be handled, but it seems to undermine the usefulness of this framework.

(2) Since the proposed framework hasn't actually been implemented, there is no evaluation of how it works or whether it helps end users choose better models and make better decisions.

**Summary Of The Paper:**

This paper proposes a framework to evaluate saliency-based explanations for neural network decisions. The framework evaluates each method along a number of different dimensions (for example, reporting whether the saliency map output is fixed or stochastic, or the degree to which the output depends on hyperparameters). The proposed framework is not actually implemented or tested in this paper.

**Summary Of The Review:**

This paper proposes a useful framework for evaluating and comparing saliency-based explanations for neural network results. However, it is entirely theoretical/speculative -- the framework has not actually been implemented or tested.

---

> ### Author Response · Authors · 2021-11-17
> **Response to Reviewer DxGz**
>
> Thank you for your thoughtful and detailed review; we appreciate you noting the usefulness of the framework for both researchers and end users.  We have incorporated several changes to the paper based on your suggestions, and provide some additional clarifications here as well:
>
> >“The methodology is not entirely transparent -- it's not clear how the models are ranked on these dimensions or how a "score" would be computed for a given dimension. The authors note that some dimensions like "perceptual correspondence" are difficult to measure and may require large-scale user studies; other dimensions may be measured in multiple different ways and different methods could give different ranking results. It's not clear how this would be handled, but it seems to undermine the usefulness of this framework.”
>
> You raise a good point. It is not obvious that there is a “best” way to measure some of the dimensions. And, in some cases, some dimensions might be dependent on external factors (e.g., semantic directness may depend on the target user group).  The goal of our framework is to provide a structured way to examine the pros and cons of various methods, not to produce an aggregate score, or to definitively rank methods as better/worse.  Rather than numerical scores, we imagine the values of each dimension to be rich descriptions that incorporate context about how they were measured. In appendix table A1, for example, we show how two different methods of examining model sensitivity can be incorporated into the descriptions of different methods, and still provide useful information for comparing across methods and understanding tradeoffs. In places where we place methods on a scale relative to each other, we also provide descriptions (e.g., next to each point in Figure 1 or in the Saliency Cards) that provide further context.
>
> >“Since the proposed framework hasn't actually been implemented, there is no evaluation of how it works or whether it helps end users choose better models and make better decisions.”
>
> While we do not conduct a large-scale user study as part of this paper, we do present initial evidence of the usefulness of our framework with end users in section 4.2 (medical case study). This section was based on 4 semi-structured interviews with radiologists to assess the ways they interpreted and compared saliency methods using our dimensions. The insights we discuss are based on these interviews.  For example, all of our interviewees described how they were accustomed to accounting for background noise when reading x-rays; as a result, they all expressed a preference for less minimal methods (e.g., preferring VG over LIME).  More generally, the dimensions of our framework provided necessary structure for comparing different methods and eliciting radiologists’ thoughts on different tradeoffs. We have now made this explicit in Sec 4.2, and have added a supplementary section that describes in detail how the interviews were conducted.
>
> These results provide initial evidence of the usefulness of our framework for comparing and choosing saliency methods. We imagine future work using the dimensions as a basis for larger-scale, formal user studies to systematically assess the pros and cons of different saliency methods across domains.
>
> >“The paper is entirely a survey/review of existing methods; it does not propose any new XAI algorithms. However, I found the proposed dimensions and comparison of the saliency methods along these dimensions very interesting and informative. I could see this framework being useful to researchers in XAI as well as end users. The paper is also very well written and clear.”
>
> Thank you for noting the potential usefulness of the framework for researchers and practitioners.  We agree that our paper does not propose a saliency method; it introduces a new evaluative framework.  As we show, the framework provides a novel and systematic way for researchers to understand the strengths, weaknesses and gaps among different methods.  Better evaluation is key to the development of new methods, by enabling researchers to characterize gaps in the existing landscape and design methods to fill them.

---

### Official Review · Reviewer_2CJF · 2021-11-02

**Correctness:** 3
**Technical Novelty And Significance:** 2
**Empirical Novelty And Significance:** 2
**Recommendation:** 3
**Confidence:** 4

**Main Review:**

The article is well written and provides a clear overview of some of the relevant works. It would have been interesting for the literature review to discuss in more detail the difference between the chosen approaches, as they calculate notably different values.

This leads to the main question from reading this article: Is it the case that the proposed taxonomy is an improvement on the state of affairs, by providing a clearer view of the pros and cons of competing approaches, or is it instead likely to obfuscate further the fact that most of those approaches compute distinctly different quantities. The paper does not provide actual evidence (for example, use case or user feedback) of this added value.

Similarly, saliency is defined as an abstraction in the introduction, but an abstraction to what is not clearly discussed, nor is the fact that there may be a disconnection between the expectations of the user and the actual quantities being calculated by different methods.

**Summary Of The Paper:**

This article proposes a review of "saliency" methods in deep neural networks and rates them according to 9 dimensions: Computational efficiency, determinism, hyperparameters, model agnosticism, input sensitivity, minimality, perceptual correspondence and semantic directness. The author's claim is that this new taxonomy provides a better way for choosing between saliency approaches for the end-user.

Note: What is meant by saliency in this paper is methods, such as Grad-CAM to determine which inputs were more prevalent in a network's decision (the word has different meanings in other sub-fields).

**Summary Of The Review:**


In summary, this article provides 1) a review of so-called saliency approaches for neural networks and 2) a new taxonomy for rating the properties of those approaches.
- The article would benefit from a more in-depth discussion of the fundamental differences between the approaches considered and what it means for the term "saliency".
- The article would benefit from some sort of evidence that the produced taxonomy is of value to the end users.

---

> ### Author Response · Authors · 2021-11-17
> **Response to Reviewer 2CJF**
>
> Thank you for your constructive comments.  We’ve updated the paper to incorporate several changes based on your comments, and provide some clarifications below as well:
>
> >“Note: What is meant by saliency in this paper is methods, such as Grad-CAM to determine which inputs were more prevalent in a network's decision (the word has different meanings in other sub-fields).”
>
> Thank you for pointing out that “saliency” can refer to different concepts across sub-fields.  In this case, we refer to saliency methods as defined in ML interpretability work: i.e., a class of attribution methods that aim to identify features in an input that were important to a trained model’s prediction.  We have further clarified this in the introduction.
>
>
> >“The article would benefit from a more in-depth discussion of the fundamental differences between the approaches considered and what it means for the term "saliency".”
>
> >“It would have been interesting for the literature review to discuss in more detail the difference between the chosen approaches, as they calculate notably different values.”
>
> We agree; each method computes saliency results differently.  Our framework is largely motivated by this fact --- our goal is to provide a language to describe and understand the fundamental differences across methods.  For example, using the dimensions of our framework to characterize methods surfaces how algorithmic differences between Integrated Gradients and LIME lead to differences in their computational efficiency, the types of models they can be applied to, and interpretation of their outputs. In Appendix Table A1, we characterize each method along each dimension, demonstrating how the framework can be used to understand their differences.
>
> We have also added more detail to Section 2 (related work) discussing and contextualizing existing categorizations of saliency methods -- for example, those that compute gradients vs. those that treat the model as a black-box and perform perturbations to the input -- and what our framework adds.
>
> >“Is it the case that the proposed taxonomy is an improvement on the state of affairs, by providing a clearer view of the pros and cons of competing approaches, or is it instead likely to obfuscate further the fact that most of those approaches compute distinctly different quantities. The paper does not provide actual evidence (for example, use case or user feedback) of this added value.”
>
> While we do not conduct a large-scale user study as part of this paper, we do present initial evidence of the usefulness of our framework with end users in section 4.2 (medical case study). This section was based on 4 semi-structured interviews with radiologists to assess the ways they interpreted and compared saliency methods using our dimensions. The insights we discuss are based on these interviews.  For example, all of our interviewees described how they were accustomed to accounting for background noise when reading x-rays; as a result, they all expressed a preference for less minimal methods (e.g., preferring VG over LIME).  More generally, the dimensions of our framework provided necessary structure for comparing different methods and eliciting radiologists’ thoughts on different tradeoffs. We have now made this explicit in Sec 4.2, and have added a supplementary section that describes in detail how the interviews were conducted.
>
> These results provide initial evidence of the usefulness of our framework for comparing and choosing saliency methods. We imagine future work using the dimensions as a basis for larger-scale, formal user studies to systematically assess the pros and cons of different saliency methods across domains.

---

### Official Review · Reviewer_JNYz · 2021-11-03

**Correctness:** 4
**Technical Novelty And Significance:** 3
**Empirical Novelty And Significance:** Not applicable
**Recommendation:** 8
**Confidence:** 5

**Details Of Ethics Concerns:**

No concerns.


**Main Review:**

Strengths
The underlying idea of framing saliency methods as abstractions that provide an approximate view of a model’s behavior is insightful. This viewpoint sidesteps the philosophical arguments about which measure is best, and instead acknowledges that different aspects of saliency methods, measured in different ways, are useful for characterizing the algorithms. One can argue about the specifics of the framework, i.e. whether the three categories are the best ones, or the nine dimensions similarly, but the underlying concept remains sound and produces a very useful analysis.

The paper format of introducing each dimension, providing its rationale, and a concrete example with a result is very effective at clearly communicating the definition and utility of the dimensions. Overall the paper is very clearly written.

The saliency cards, Fig. 4, are an excellent snapshot of each saliency algorithm. These types of cards have proved to be useful in other domains, as the authors state, and should be quite useful for user-facing algorithms such as saliency.
The analysis summary in table 1A is a compelling example of how the framework can comparatively reveal strengths, weaknesses and gaps in particular algorithms.

It is more than a survey paper, through its careful definition of the nine dimensions, many of which are new to this work. The concepts of perceptual correspondence and semantic directness seem particularly unique; the latter attempts to quantify the inherently squishy notion of the ease of user understanding of a method’s results.

Weaknesses
All of the highlighted saliency methods are from 2017 or earlier, except one from 2019. While it is reasonable to focus on those that have established themselves over time as impactful, it would be helpful to incorporate a few more recent methods to ensure that they still fit into the framework.

While the paper should be quite useful for researchers and practitioners of saliency algorithms, it does not present novel work to build upon. Some may find this insufficient for ICLR.


**Summary Of The Paper:**

The paper describes a new framework for characterizing and analyzing saliency methods. Nine dimensions are defined, grouped into three high-level categories, with some form of quantification in each dimension to enable comparative measures. The dimensions are used to formulate saliency scorecards, concise 1-page summaries of the saliency attributes of a particular algorithm that should help a user determine which algorithm is best for their needs. A set of ten recent algorithms are analyzed and compared using the framework, revealing that some have gaps in their assessments that could be address with future work. Compelling, illustrative examples are used throughout to convey understanding of the dimensions and the differences between them.

**Summary Of The Review:**

Papers like this one tend to be rejected from highly selective conferences, but they can be very useful and impactful for researchers and practitioners alike in popular areas, such as saliency algorithms. This paper is thoughtful, well positioned with respect to previous attempts to characterize saliency algorithms, and should be an excellent resource for the saliency research community.

---

> ### Author Response · Authors · 2021-11-17
> **Response to Reviewer JNYz**
>
> Thank you for this thoughtful review!  We are glad you found the framing of the paper insightful and the characterization of each dimension effective.
>
> Based on your suggestion, we incorporated a few more recent methods:
>
> >“All of the highlighted saliency methods are from 2017 or earlier, except one from 2019. While it is reasonable to focus on those that have established themselves over time as impactful, it would be helpful to incorporate a few more recent methods to ensure that they still fit into the framework.”
>
> As you note, we primarily focused on methods that are well-established and widely-used.  However, we agree that demonstrating the framework’s applicability to newer methods would strengthen our evaluation.  We have added two more methods to the paper: XRAI (Kapishnikov et al. 2019), and RISE (Petsiuk et al., 2018).  We choose these methods since they are fairly widely-used and cited, and they have open-source implementations.  We have incorporated them throughout the paper and characterized them across each dimension in Appendix Table A1.

---

### Decision · Program_Chairs · 2022-01-20

**Decision:**

Reject

**Comment:**

The reviews end up split. The most positive reviewer notes that the paper may be useful broadly to researchers and practitioners. That is the main promise of the work. However as another reviewer points out, the authors fall short of convincingly explaining how the said practitioners and researchers would benefit from the work. And, also as noted in a review, the paper does not quite go deep enough into the discussion of what saliency means for different methods analyzed.
Contrary to the positive reviewers comment, I do not think that a paper must provide "novel work to built upon", but for a paper that doesn't, there is a significant threshold on how convincing it must be regarding the potential impact of the work. I think the paper in its current state, even after the rebuttal/revision, does not meet this threshold.